# Neural Bases of Language Recovery After Stroke Can Only Be Fully Understood Through Longitudinal Studies of Individuals

**DOI:** 10.3390/brainsci15080790

**Published:** 2025-07-25

**Authors:** Argye E. Hillis

**Affiliations:** 1Department of Neurology, Johns Hopkins School of Medicine, Baltimore, MD 21287, USA; argye@jhmi.edu; Tel.: +1-(443)-287-4610; Fax: +1-(410)-614-9807; 2Department of Cognitive Science, Johns Hopkins University, Baltimore, MD 21218, USA; 3Departments of Physical Medicine and Rehabilitation, Johns Hopkins School of Medicine, Baltimore, MD 21287, USA

**Keywords:** aphasia, recovery, ischemic stroke, functional imaging, methodology

## Abstract

Despite decades of intense interest and investment in cognitive science, there remains a not only incomplete but also highly inconsistent body of evidence regarding how adult brains recover from even the most focal injuries associated with stroke. In this paper, I provide a broad narrative review of the studies of post-stroke aphasia recovery that have sought to identify the mechanisms of language recovery through longitudinal functional imaging. I start with studies that used functional imaging in groups of neurotypical individuals that have revealed areas of the brain that are reliably activated by language tasks and are functionally connected, referred to here as the “language network.” I then review group studies in which functional imaging data were averaged across groups of people with post-stroke aphasia to characterize the neurobiology of recovery. These group studies of post-stroke aphasia have yielded very different results and have led to conflicting conclusions. Subsequently, I examine results of studies of single subjects that have employed longitudinal functional imaging to identify mechanisms of language recovery. Together, these single subject studies make it clear that mechanisms of neural recovery are far from uniform, even in people with very similar lesions and time since stroke. On this basis, I argue that it is not justifiable to average functional imaging data across individuals with post-stroke aphasia to draw meaningful insights into how brain networks change to support language recovery. Each individual’s brain networks change over time, but in divergent ways that depend on the extent of disruption to the normal language network, interventions to facilitate recovery, the health of the intact brain, and other variables yet to be identified. While averaging results across participants with post-stroke aphasia might be able to identify certain changes in the networks that are correlated with specific language gains, uncovering the range of mechanisms and dynamics of language recovery after stroke requires longitudinal imaging of individuals.

## 1. Introduction

Here I provide a narrative review of functional imaging studies that pertain to language recovery after stroke, selected based on their relevance to the topic of neural mechanisms of aphasia recovery. First, I review results of functional imaging of language in neurotypical people that have revealed a reliable “language network” in the left frontal, temporal, and parietal cortex. These studies that have averaged data from healthy controls have yielded convergent, highly valuable insights into areas of the brain that work together to support language. Then I review longitudinal functional imaging studies of recovery of language *after stroke* that have reported average results across a group or groups. This review reveals contrasting results with respect to both changes in cortical activation and changes in connectivity associated with recovery across studies. I argue that such contrasting results are inevitable, as studies report average results across groups of people with different volumes and locations of strokes (as well as different demographics). I then review longitudinal functional imaging studies of individuals with post-stroke aphasia (PSA) that illustrate the many distinct ways brain networks can change with recovery of language, depending on not only time post-stroke, but also lesion characteristics, patient characteristics, and interventions.

## 2. Functional Imaging Studies of Language in People Without Brain Lesions

Functional imaging studies of heathy individuals have reliably revealed a network or cortical regions that are activated in nearly every language task. Although some designs have revealed distinct areas activated more during one language task versus another (e.g., naming actions versus objects) nearly all language tasks engage a network of regions of left hemisphere when contrasted with low level baselines that are primarily attentional or perceptual. Nodes of this “language network” nearly always include left (more than right) posterior inferior frontal gyrus (pIFG; often referred to as “Broca’s area”), posterior superior temporal cortex (pSTG, often referred to as “Wernicke’s area”), middle temporal gyrus (MTG), inferior temporal gyrus or fusiform gyrus (FuG), dorsolateral prefrontal cortex (DLPFC), supramarginal gyrus (SMG) and angular gyrus (AG). This language network has been revealed by functional MRI (fMRI) studies across tasks of word retrieval [1], passive viewing and listening to discourse [2], comprehension [3], and reading [4], as well as by PET studies (see [5] for review). Unsurprisingly, tasks with visual stimuli also activate occipital areas not central to the language network, unless compared to a baseline with similar visual demands.

These brain regions that comprise the “language network” are nodes of a network revealed by task-free (“resting state”) fMRI, as indicated by highly correlated Blood Oxygenation Level Dependent (BOLD) activation (referred to as “high connectivity”) [6]. However, superior frontal cortex is also among the nodes with high connectivity with language network areas [6].

Of note, essentially all functional imaging studies of language also show activation of at least some right hemisphere regions homologous to the language network, although typically lower than seen in the left hemisphere [5,7,8]. These studies indicate that right hemisphere homologues to language network might have a supportive role in language processing.

Functional imaging studies of neurotypical controls that have formed our understanding of the brain network underlying language have largely been conducted in group studies, showing significant activation for the group. However, individual neurotypical controls reliably show the same areas of activation associated with language as do groups, although longer imaging studies (generating more data) are often required to obtain significant results [9].

## 3. Functional Imaging Studies of Recovery in Groups of People with PSA

Many clinical investigations are launched with the express aim to identify the areas of the brain activated by language when part of the language network is damaged by stroke. The goal of these studies is to use groups of patients averaged together to identify the neural basis of recovery from PSA. Do healthy regions in the left hemisphere or in the right hemisphere “take over” the functions of the damaged nodes of the network? Answers to such questions have an important clinical as well as scientific impact. For example, should rehabilitation of aphasia focus on tasks or interventions (e.g., non-invasive brain stimulation) that recruit the right hemisphere?

Unfortunately, unlike broader characterization of healthy function, there are seemingly conflicting answers to these critical questions in groups of patients with lesions. A summary of results across studies are reported in Table 1. An early study of a group of seven participants who had substantially recovered language after aphasia by the five months after stroke revealed significantly increased activation of right hemisphere areas, and non-significant increase in left hemisphere areas, compared to controls, associated with a lexical-semantic task [10]. Participants with post-stroke aphasia who recovered best showed bilateral activation of language network regions and their homologues. In contrast, in a study of four participants with PSA and four controls, positive activation associated with picture identification was observed predominantly in perilesional areas (along with negative activation in the right hemisphere). Both PSA and control groups activated left frontal and temporal language areas, but in those with PSA, the frontal region activation was spread over a larger perilesional area. They concluded that people recovering from post-stroke aphasia use functional perilesional areas to perform language functions [11]. However, a study of 82 people with PSA and 82 controls showed no evidence of perilesional activation associated with naming or semantic decision in PSA recovery [12]. Instead, they proposed that aphasia recovery depends on “normalization of language network” (neurotypical patterns of activation in left inferior frontal, prefrontal, inferior parietal, and temporal regions) and possibly recruitment of alternative areas of the cortex, independent of the distance from the lesion. Very recently, the same group reported a large study of 76 people with chronic PSA and 69 neurotypical controls engaged in a semantic decision task. They showed, on average, right hemisphere activation associated with the language task was higher in participants with PSA than in controls. Furthermore, greater right hemisphere activation was associated with higher education, younger age, and left-handedness, but (in contrast to previous studies) was not associated with lesion size or longer time since onset of stroke. Finally, only those with PSA activated right dorsal inferior frontal gyrus during this task [13].

Saur and colleagues studied 14 people with PSA acutely (0–4 days), subacutely (2–3 weeks), and chronically (4–12 months) with a sentence comprehension task [3]. They report minimal activation acutely, mostly increased right hemisphere activation subacutely, and finally normalization of left hemisphere language network chronically. But another group who studied 17 people with PSA at 2, 6, 17, 26, and 52 weeks post-stroke reported mostly increased left temporal and left cerebellar activation during a semantic fluency task over time [14]. There was also a shift toward left frontal lateralization. They also observed temporary compensatory activation of the right hemisphere but concluded it was not important in recovery, as recovery was driven by reactivation of left fronto-temporal regions. However, a separate study of eight people with mild PSA and left IFG lesions showed increased activation of right frontal and left cerebellum associated with learning effects on a word-retrieval task [15].

Some studies have divided PSA into separate groups to determine if there are distinct mechanisms of recovery or studied people with PSA longitudinally (as summarized in Table 2). One study of nine individuals who had recovered well from PSA showed predominantly left hemisphere activation during language, while 18 people who had limited recovery showed mostly right hemisphere activation. They concluded that a right hemisphere “shift” is an ineffective mechanism in language recovery [16]. Another evaluated 17 people with mostly frontal strokes and 17 with temporoparietal strokes as the cause of PSA. They found that both groups showed reduced activation during language, including in areas far from the lesion (diaschisis) followed by reactivation of the normal language network (i.e., resolution of diaschisis) by 2–3 weeks. By 6 months, both groups showed increased activity of perilesional cortex and reorganization of left temporal language areas in the chronic phase. They found activation of right hemisphere homologues in the group with frontal but not temporoparietal lesions [17]. An investigation of resting state connectivity before and after treatment in a trial of lexical therapy demonstrated differences between groups dichotomized by high improvement with therapy versus low improvement with therapy [18]. Those who showed good improvement showed an increase in average connectivity between left fusiform gyrus (an area shown to be critical for picture naming) and other regions in the language network, as well as increased connectivity between left fusiform cortex and right hemisphere homologues after treatment. Those who showed no or poor improvement with treatment showed decreased connectivity between left fusiform gyrus and left language network and right hemisphere homologues.

A recent study used functional near-infrared spectroscopy (fNIRS) both to guide transcranial magnetic stimulation (TMS) and to evaluate changes in language activation before and after an intensive language therapy in eight people with chronic PSA. They found that average changes in activation depended on the treatment group [19]. Participants who received low frequency stimulation to the right inferior frontal gyrus (IFG) showed reduced activation in both hemispheres, but stronger activation in left than right hemisphere with improvement in language after treatment. In contrast, those who received high frequency stimulation to right IFG showed increased right relative to left hemisphere activation with improved language after therapy. Hartwigsen and Saur recently provided a comprehensive review of fMRI and PET studies of aphasia recovery with various treatments (beyond the scope of this review), showing a variety of changes in average activation associated with treatment success across studies, including (1) upregulation of right hemisphere regions; (2) downregulation of right hemisphere regions; (3) upregulation of bilateral regions; (4) upregulation of perilesional or spared left hemisphere regions; (5) downregulation of perilesional or spared left hemisphere regions [20].

This review of the functional imaging group studies of PSA recovery clearly reveals conflicting results across studies. This conclusion is not surprising, given that the groups studied were variable in terms of stroke characteristics and given that different tasks were evaluated with fMRI. Moreover, each study averaged the results across people with aphasia who have different lesions, and often different times post-stroke, different degrees of recovery, different initial deficits. Some studies have shown that the average site of activation does depend on the time post-stroke [3,17]; others have shown that the average site of activation depends on lesion location [17], or degree of recovery [16]. Others found that average changes in activation depend on age, education, and left-handedness [12]. Further, average changes differ depending on interventions for aphasia [21].

Hence, while it is justifiable to average results of functional imaging of language in neurotypical controls to increase power, given that studies have shown a fairly high degree of homogeneity across neurotypical individuals engaged in the same task, it is not justifiable to average functional imaging across individuals after stroke. As we review below, individuals with PSA vary widely in what areas (and even what hemisphere) there is increased or decreased activation associated with any given language task. The average result for any given group may not be representative of any single participant in the study, and is certainly not generalizable to other people with PSA.

Group studies of aphasia recovery using functional imaging may provide insights into changes in activation or connectivity that correlate with recovery. For example, a longitudinal fMRI study of PSA showed that in the subacute stage of recovery, right supplementary motor area activation correlated with language improvement [3]. A recent fNIRS study of 20 people with PSA showed that stronger resting state connectivity within right hemisphere and between hemispheres significantly correlated with higher aphasia quotient and better naming on standardized tests [22]. These types of correlations between specific activations or connections would represent the group. However, these results might not generalize to other groups of people with PSA. It would be interesting to test such correlations (using the same tasks and same imaging) in groups of patients with similar lesions and the same stage of recovery and in groups of patients with different lesions, or different stages of recovery to determine the reliability of the findings. A longitudinal study of fMRI in 17 people with aphasia due to frontal or temporoparietal stroke revealed that correlations between (1) language improvement and (2) task-related functional interactions between the language network and the multiple-demand network depended on lesion location and changed over time [23]. A key point is that studies that evaluate correlations between improvement in scores and activation or connectivity might provide insights into the neural bases of recovery across individuals (likely only those with specific lesions and stages of stroke), whereas studies summarized in Table 1 do not provide clear insights about the recovery of any individual in the group or anyone else with PSA.

## 4. Functional Imaging Studies of Recovery in Individuals with PSA

Given the conflicting results obtained from group studies regarding changes in activation with recovery from PSA, depending on a variety of lesion and patient variables, the only clear solution is to evaluate changes in series of individuals of all types (ages, sexes, times post-stroke, variety of lesion volumes and locations) to unveil the range of changes in the brain that support language recovery after stroke. In this section I review some of the published studies of functional imaging of individuals who have recovered from PSA, as well as longitudinal studies of individuals during PSA recovery. Many are from my own lab, as I have available the original images (not previously published) to illustrate distinct mechanisms of recovery. The studies certainly show heterogeneous results, but do not draw conclusions meant to generalize across people with PSA. Rather, the aim is to show the range of mechanisms of recovery.

### 4.1. Task-Related fMRI Studies of Recovery in Individuals with PSA

An early study evaluated fMRI in two patients with residual non-fluent PSA who each received two different interventions. One person showed improvement on one treatment (focusing on “intention”) but not on the other treatment (focusing on “attention”). This individual showed a shift in activation to right presupplementary motor area and the right lateral frontal lobe after treatment in a task of generating names given a category such as “birds”. The second patient showed improvement on both treatments. Before treatment this individual showed right hemisphere activation during language that persisted after treatment, but also showed increased activation in the left posterior peri-sylvian cortex during the naming within category task [24].

Another early fMRI study included seven people with left middle cerebral artery ischemic stroke infarction who had partially recovered comprehension after initial diagnosis of global aphasia. The most common areas of activation across individuals associated with a semantic task compared to a lexical task were **left** extra-sylvian posterior temporal and the **right** posterior parietal cortex. The semantic task was a category decision task (animal vs. non-animal natural item), and the lexical task was a lexical decision task with reversed words serving as the non-words. Those with the best recovery of language comprehension showed activations in regions that were also activated in several neurotypical controls [17,25].

The same group reported results of fMRI of word processing from two individuals with PSA who had partially recovered. One showed left perilesional inferior and middle frontal activation extending to parietal and left superior temporal gyrus as well as left thalamus, precuneus, and precentral gyrus in a task of lexical decision compared to a task of discriminating reversed words from sounds with a dynamic amplitude envelope but without spectral frequency shifts. The second person showed a single highly significant cluster of activation in left posterior superior, middle and inferior temporal gyrus associated with the same task comparison. Activation in parts of Wernicke’s area (posterior superior temporal gyrus) was the only overlap in activation associated with “conceptual-semantic word processing” (greater activation during lexical decision). No right hemisphere regions were significantly activated [21,26].

On the other hand, an fMRI study of word generation in three people with PSA who had partially recovered language, compared to six neurotypical controls, revealed no activation in left inferior frontal gyrus (whether or not there was a lesion in the left frontal lobe), but some right hemisphere activation in two people with PSA, in areas not activated by controls. The neurotypical controls, as expected, showed activations in left frontal, temporal, parietal and occipital regions [27].

One study obtained both diffusion-weighted imaging to reveal the infarct and perfusion weighted imaging to reveal the area of dysfunctional but salvageable tissue beyond the infarct in two people with acute PSA. They showed that successful restoration of blood flow resulted in task-specific activation in the perilesional tissue within the previous area of salvageable tissue on perfusion weighted imaging [28]. The task was a sentence completion task using the correct inflection of the given verb. These results may not reflect any “reorganization”, but recovery through reperfusion.

A later study of individuals with multimodal imaging acutely and at subsequent time points also demonstrated distinct mechanisms of recovery, including (a) reperfusion; (b) recovery from diaschisis; (c) “reorganization”, whereby undamaged regions show increased activation during recovery [26,29].” Reorganization” may actually be a process of relying more heavily on previously present but “supportive” or underused networks (e.g., regions homologous to the left hemisphere language network) [27,30]. While a previous group study indicated that, on average, diaschisis occurs acutely in PSA [3], this case series found that acute diaschisis was far from universal [29]. One individual with PSA due to a left thalamic lesion showed clear evidence of recovery from diaschisis. This individual showed reduced activation in the left hemisphere language cortex but activation of the right hemisphere homologous network at one month post-stroke (subacutely) associated with picture naming (vs. saying “scrambled” in response to a scrambled picture), followed by reactivation of the left language network at 3 months post-stroke (Figure 1).

However, other individuals showed activation of perilesional tissue in the language network even acutely as shown below. In this series of individual task-related fMRI studies at several points in the first year of stroke, the degree to which language activated normal left hemisphere language network depended not only on time from stroke onset, but also on the language task, accuracy of performance on the language task, the volume and location of the stroke.

With regard to the effect of accuracy of performance, two women of similar ages with very similar infarcts involving left posterior inferior frontal cortex (“Broca’s area”), showed quite different patterns of activation during a spelling task (identifying the missing letter in a word vs. identifying the letter with a different case in a word) at Day 3. While the participant with 100% accuracy showed bilateral (right more than left) activation in areas similar to controls, the participant with 80% accuracy showed only left hemisphere activation as shown in Figure 2. These data raise doubts that right hemisphere homologous activation is detrimental, since it was associated with flawless performance [26,29].

Likewise, another task-related fMRI study from our group involved a man who recovered well from post-stroke global aphasia after a massive left middle cerebral artery stroke. Three years post-stroke he scored in the normal range on the Western Aphasia Battery [31]. His reading was also 100% accurate and fluent. During a reading task (reading aloud words vs. saying “skip” to letter strings), he showed only right hemisphere activation of fusiform cortex (Figure 3), also confirming that right homologous activation can be positive, and associated with flawless performance, even at the chronic stage of stroke [29,32].

To illustrate the effect of language task, the woman described earlier who showed only left occipital activation with spelling showed bilateral activation and more left perilesional activation during reading (silent reading of words versus viewing of scrambled letter strings) (Figure 4). She read the words with 100% accuracy outside of the scanner that day. Her activation with reading was essentially the normal pattern seen in neurotypical controls [30]. While it might be tempting to conclude that the minimal activation of perilesional tissue during **spelling** was due to diaschisis, there was no evidence of diaschisis during the **reading** task the same day [26,29].

Another longitudinal fMRI study from our group, involving a third woman with PSA (with left occipital and splenial lesion) also showed distinct but overlapping areas of activation for reading, spelling, and naming at four time points from acute to 1 year after stroke. Reading, spelling, and overt naming tasks were the same as those described in the three paragraphs above, in other studies from our lab. The areas where changes in activation were associated with improvement in each task (Time 4–Time 1), were also distinct, but adjacent, as shown in Figure 5. Red bars show upregulation of activation in the left inferior frontal gyrus and mid-fusiform gyrus (FG) reading and spelling. The green bars show upregulation of activation in left posterior fusiform with recovery during reading and naming [31,33].

### 4.2. Functional Connectivity MRI Studies of Individuals with PSA

Both task-related and resting state (or task-free) fMRI studies of PSA have also frequently reported averaged results across participants with different lesions, as described in Section 3 above. However, studies that report results of changes in connectivity show distinct changes in connectivity over time, even in individuals with similar lesions.

For example, one longitudinal study of four individuals, all with aphasia due to stroke in the left posterior cerebral artery territory reported different patterns of activation associated with an overt naming of pictured objects task at each of four times in the first year after stroke [32,34]. All of the individuals showed activation in left language network regions and right hemisphere homologues, but with different degrees of activation at each time point. The best recovery of naming was associated with increased balance in activation across nodes of the left hemisphere language network (including left inferior frontal gyrus, posterior temporal gyrus, angular gyrus, supramarginal gyrus and fusiform gyrus) and their right hemisphere homologues, again raising doubts that homologous activation is generally detrimental. This study also included connectivity analysis of the longitudinal fMRI in the four individuals. Three individuals showed improvement in naming accuracy from the acute to the chronic stage, which was associated in each case with increased connectivity within and between left hemisphere language regions, and their right hemisphere homologues (e.g., P1, Figure 6). In contrast, the individual who showed a worsening naming deficit (P3) showed weak and decreasing connectivity within and between left and right hemisphere language network areas, as shown in Figure 6.

One study that reported both group and individual results from longitudinal resting state connectivity fMRI emphasized the role of network modularity in language and other networks in functional recovery [35]. They studied 107 people with left or right hemisphere stroke and found that the degree of within-network integration (within and across hemispheres) and across-network segregation between networks (visual, default-mode, somato-motor, auditory, salience, dorsal attention, ventral attention, cingulo-opercular, fronto-parietal) was significantly reduced at 2 weeks (n = 107), but increased at 3 months (n = 85), and 12 months (n = 67) post-stroke. Within-network connectivity (and across-network segregation (“modularity”) correlated with recovery of language, spatial memory, and attention, but not with motor or visual function. Detailed analysis of an individual with severe aphasia due to a left temporoparietal stroke showed loss of connectivity between frontal and temporoparietal regions within and across hemispheres at 2 weeks and substantial recovery of both language performance and modularity in multiple networks by three months post-stroke.

### 4.3. Functional Near-Infrared Spectroscopy (fNIRS) Studies of Recovery in Individuals with PSA

fNIRS is another modality of functional imaging, based on the Blood Oxygen Level Dependency (BOLD) effect, like fMRI, which has been used to study both resting state functional connectivity and changes in activation with aphasia recovery, although only a few studies have been published to date. Results are typically reported as change in oxygenated hemoglobin concentration [HbO] relative to blood flow, as well as deoxygenated hemoglobin concentration relative to blood flow. Similar differences in mechanisms of recovery across individuals have been identified as reported in fMRI studies above. For example, Figure 7 and Figure 8 show contrasting changes from before to after treatment in a randomized sham-controlled trial of computer-delivered lexical therapy (with transcranial direct current stimulation or sham) in two participants with subacute aphasia (<3 months post-stroke at start of trial). The individual with a small subcortical stroke showed reduced activation ([HbO], in red) in left hemisphere language network before treatment, when naming was severely impaired, but increased activation in left hemisphere language cortex (especially superior and middle temporal gyrus) when naming had improved. The task was overt naming of pictured objects compared to saying “skip” or “no” in response to a scrambled picture. The observed change was likely due to recovery from diaschisis, since there was no damage to the left hemisphere language cortex (similar to the results shown in Figure 1 with fMRI) [22,34].

In contrast, a participant in the same randomized trial who had a large left MCA stroke with damage to the left hemisphere language cortex showed poor subacute recovery but did produce more intelligible words after treatment. This individual with global aphasia two months post-stroke that involved the language network (Figure 8, top panel) showed increased perilesional left language cortex activation in the overt naming task described above (especially in left temporal and parietal cortex, Figure 8 (left lower panel) but also increased activation in some right hemisphere homologues (right lower panel) compared to pre-treatment (middle panel) and also compared to controls, as shown in Figure 7, top panel) with increased intelligible word production [22].

### 4.4. Transcranial Magnetic Stimulation (TMS) with Functional Imaging of Recovery in Individuals with PSA

Studies combining functional imaging, behavioral, and transcranial magnetic stimulation in individuals with aphasia have also provided enlightening results. One woman showed improvement in naming after inhibitory TMS to the right IFG pars triangularis, maintained two months later. fMRI overt picture naming compared to pattern viewing showed reduced activation at the site of TMS target (without increased activation of the left hemisphere homolog). Three months after her treatment with TMS, she sustained a right hemisphere stroke that caused worsened aphasia, indicating that while right pars triangularis activation was detrimental (such that inhibition improved naming), other right hemisphere areas had been supporting language recovery (such that damage caused worse language) [23].

Another study reported results from 11 individuals with PSA. Using PET, they found activation of inferior frontal gyrus (IFG) associated with language (deciding whether a spoken verb matched a drawn object) in the left hemisphere only in three participants and in both hemispheres in eight participants. Five of the eight individuals who showed right IFG (and none who had only left IFG activation) showed increased errors or latency in a semantic task with inhibitory TMS to the right IFG, indicating an essential role of right IFG in language function in five of 11 participants [36].

## 5. Discussion

Here I have illustrated the divergent results of studies of language recovery in PSA that have used data averaged across groups, which have yielded conflicting conclusions. I have proposed an alternative approach to identifying the various ways the brain adapts to a sudden, focal lesion to the “language network” to recover language, using longitudinal single subject functional imaging. These single subject case series unveil distinct patterns of recovery across individuals, which may depend on the location and extent of stroke, time since stroke, performance accuracy, and perhaps various demographic factors. The conclusions drawn from averaged group results are necessarily limited to the group studied, and have minimal contributions to understanding the recovery of any individual in the group, much less to the understanding recovery of individuals with different lesions and abilities.

Here are also some common findings from group studies of aphasia recovery. For example, most show that the best recovery from aphasia is associated with the extent to which the individual can recruit the normal “language network” in the left hemisphere. But this conclusion seems relatively obvious. Unfortunately, many people with large left middle cerebral artery strokes are unable to recruit the normal left language network as it has been obliterated by the stroke. A recent review of fMRI studies of aphasia recovery reported that most often there is a global network breakdown acutely after stroke, followed by normalization of left hemisphere language networks with language recovery. But importantly, the authors noted that individual characteristics were associated with increased right hemisphere activation and sometimes activation of bilateral domain-general regions with recovery [37]. (see also [38]). It is important to understand how individuals with destruction of the left hemisphere language network (who cannot show normalization of this network) sometimes recover language. Combining their results with those with strokes that preserve the language network and reporting group averages will not help in this endeavor.

## 6. Conclusions

While group studies of aphasia recovery have yielded important insights into the dynamic changes in activation and connectivity between the language network regions and their right hemisphere homologues, longitudinal functional imaging studies of individuals are essential to discover the various ways the neural networks change with language recovery (or language decline) after stroke. To make progress in this area it is essential to abandon the common practice of averaging functional imaging data across participants with distinct lesions and divergence in other variables that affect recovery. Case series of functional imaging of recovery should systematically investigate the range of changes in the neural networks that support recovery of a variety of language tasks, as illustrated above for a small number of tasks [25,26,27,34].

What is needed is a collaboration of investigators who agree to bank data on a set of agreed upon imaging protocols as well as behavioral data, from individuals with PSA. Such large data sets have proven to be exceptionally useful for uncovering changes in the brain related to changes in behavior, such as the multitude of studies generated from the Alzheimer’s Disease Neuroimaging Initiative (https://adni.loni.usc.edu/). The Collaboration of Aphasia Trialists, an international network of aphasia investigators from numerous disciplines and from across more than 50 countries, has begun to discuss such an initiative focused on aphasia recovery (https://www.aphasiatrials.org/). Such a data set would provide the opportunity to use innovative statistics and artificial intelligence to describe and visualize the variations in recovery trajectories in the brain and in behavior, and how they are linked. For example, lasagna plots [39] can be used to visualize changes in language tasks across individuals over time. Although I would avoid averaging data across patients, with sufficient number of participants, non-parametric statistics can be used, for example, to evaluate associations between frequency or degree of activation or connectivity in particular areas and improvement on specific language tasks. But undoubtedly, biostatisticians will develop more sophisticated and clever ways to describe the variations in, and test hypotheses about, how brain activation changes in association with recovery of language.

## Figures and Tables

**Figure 1 brainsci-15-00790-f001:**
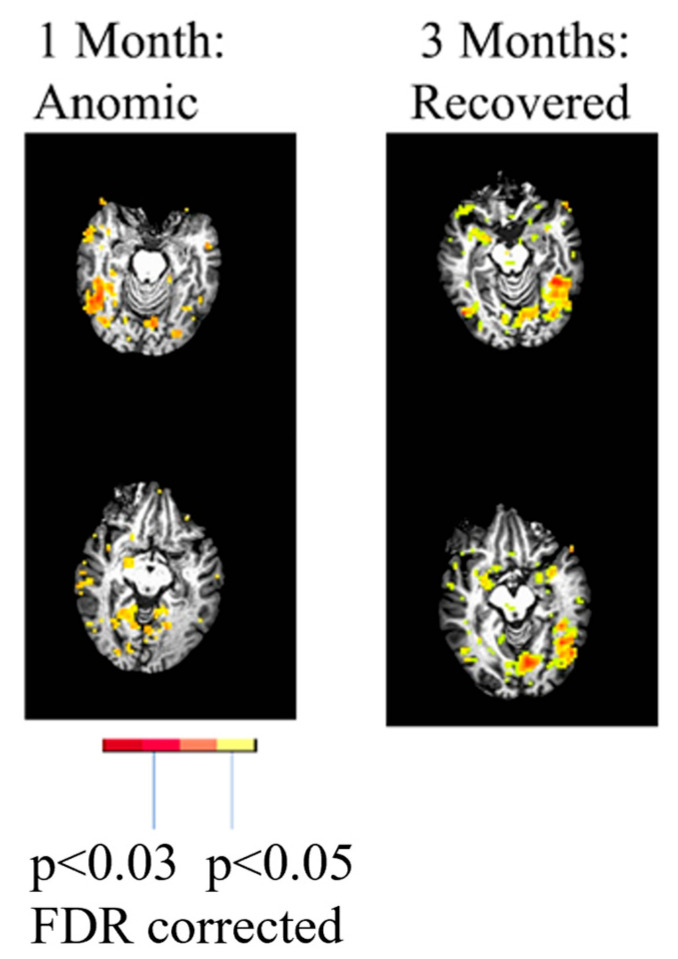
fMRI study of object naming showed minimal activation in the intact left temporal cortex at one month but activation of the homologous right temporal cortex, when he had slow but accurate naming (**left**). He showed normal activation of left temporal cortex at 3 months, when his naming had recovered in terms of latency (**right**).

**Figure 2 brainsci-15-00790-f002:**
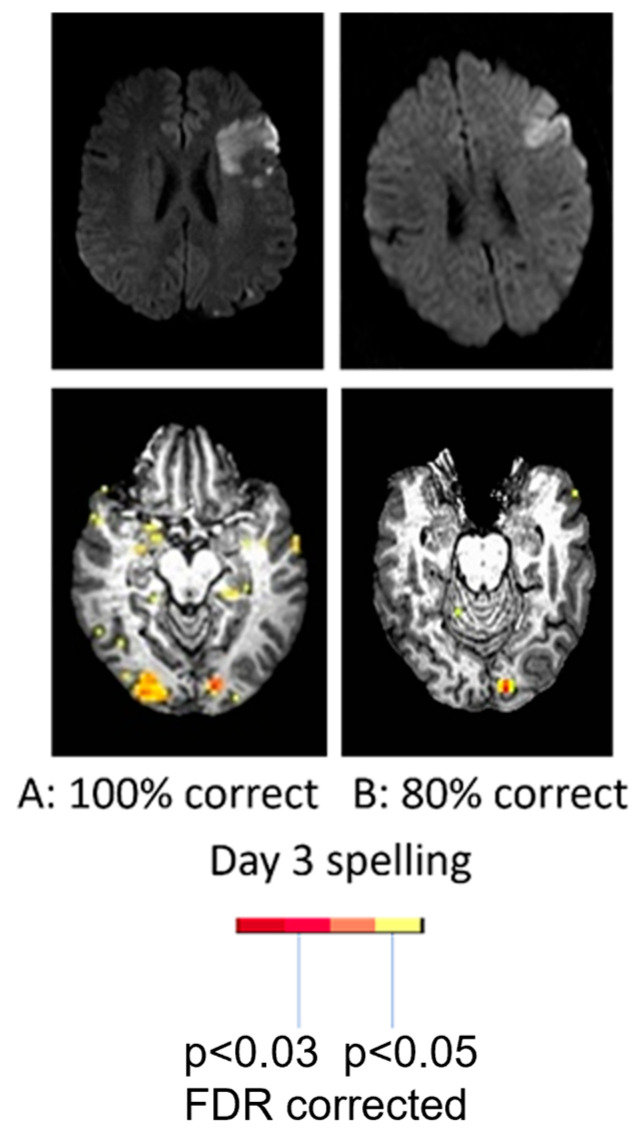
(**Top**) Diffusion-weighted images showing acute infarct in similar areas of left IFG. (**Lower panel**) Bilateral occipital activation and temporal activation during picture naming in the woman with accurate naming (**left**); only left occipital activation during picture naming in the woman with impaired naming (**right**).

**Figure 3 brainsci-15-00790-f003:**
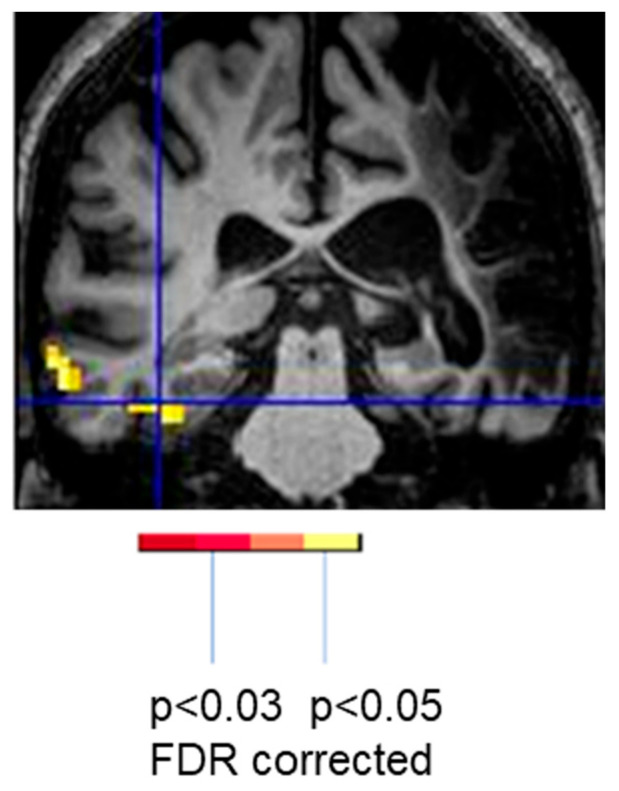
Reading-related activation in the right (but not left) fusiform cortex associated with flawless reading 3 years post-stroke in a right-handed man who had largely recovered from a massive left hemisphere stroke and global aphasia.

**Figure 4 brainsci-15-00790-f004:**
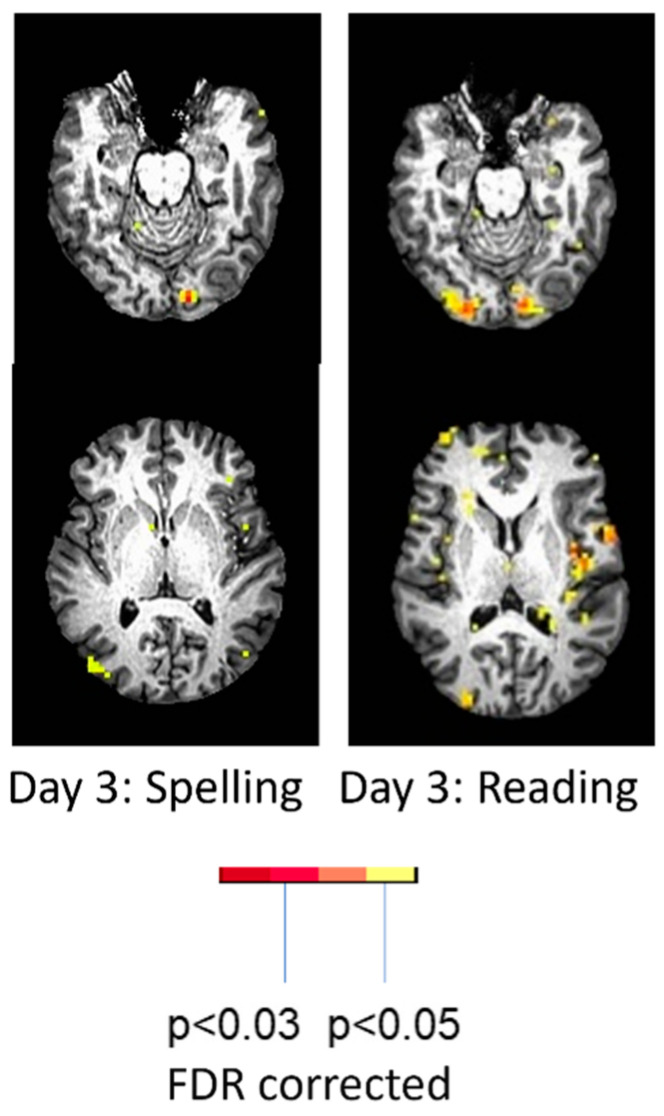
Task-related fMRI activation at Day 3 (acutely) post-stroke in the same woman associated with spelling (80% correct; **left**) and reading (100% correct; **right**).

**Figure 5 brainsci-15-00790-f005:**
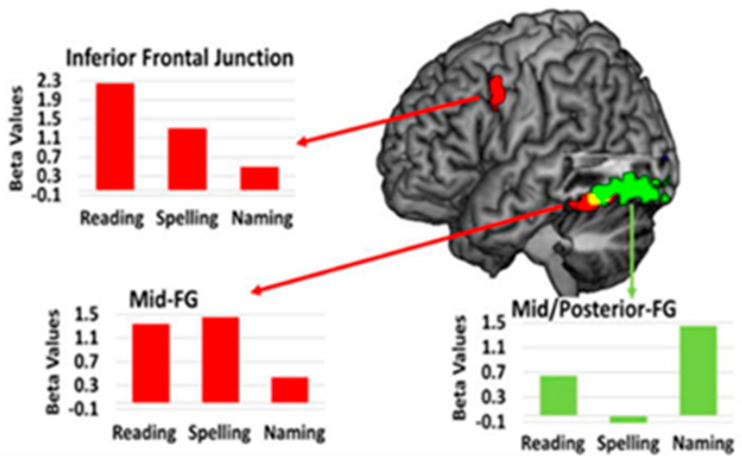
Sites of change in task-related activation at 12 months compared to acute in a woman who showed excellent recovery of reading, naming, and spelling after a left posterior cerebral artery stroke.

**Figure 6 brainsci-15-00790-f006:**
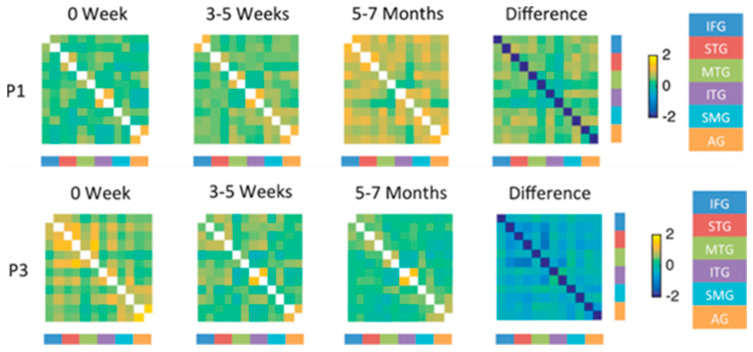
fMRI connectivity between language network areas and their right hemisphere homologues (left and right hemisphere represented by adjacent bars and columns in each network). The right hand color bar shows each area: IFG = inferior frontal gyrus; STG = superior temporal gyrus; MTG = middle temporal gyrus; ITG = inferior temporal gyrus; SMG = supramarginal gyrus; AG = angular gyrus. Yellow colors in each matrix show increased connectivity; blue colors in each metric represent decreased connectivity.

**Figure 7 brainsci-15-00790-f007:**
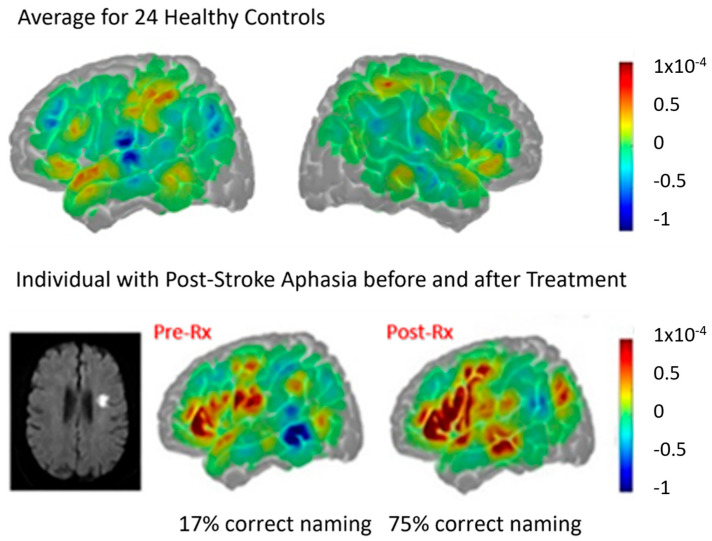
fNIRS showing activation ([HbO] in red), associated with overt naming, for 24 neurotypical controls (**top**), and one individual with small subcortical stroke at 2 months post-stroke before participating in a therapy trial (**lower**, **center panel**) and 2 months later after the trial (**lower**, **right panel**) when naming had improved.

**Figure 8 brainsci-15-00790-f008:**
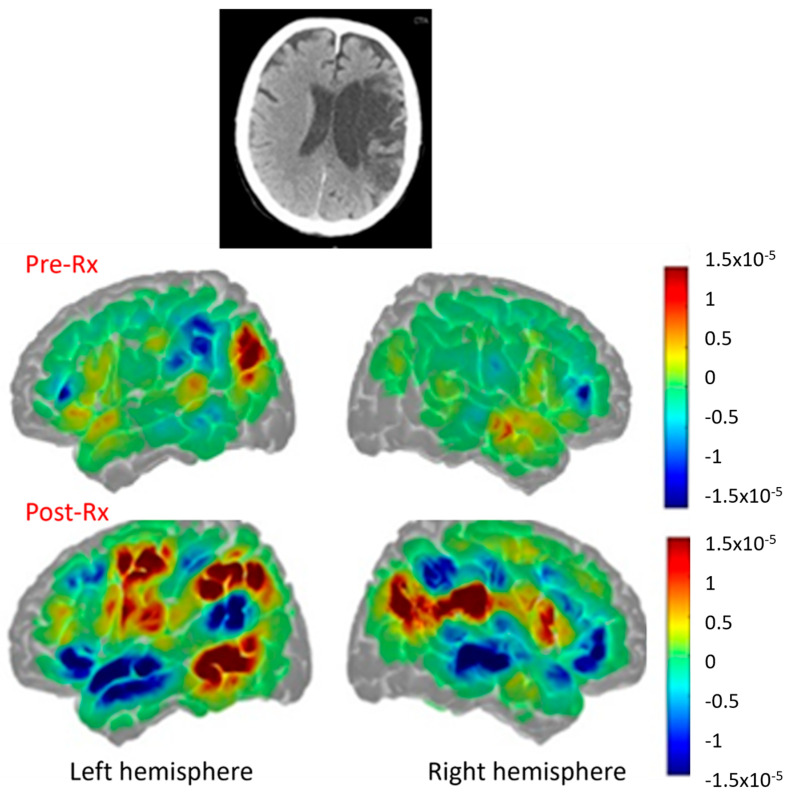
(**Top panel**) Large left middle cerebral artery infarct. (**Middle panel**) Pre-treatment activation in red [HbO]. (**Lower panel**) Post-treatment activation with naming in red, with substantial increases in left frontal, parietal, and temporal cortex and right frontal and parietal cortex.

**Table 1 brainsci-15-00790-t001:** Group studies of PSA recovery.

Reference	Number (N)	Time Since Stroke	Task	Main Results
[10]	7 PSA	5 months	Lexical–semantic	Increased right hemisphere activation, but bilateral language network activation associated with good recovery
[11]	4 PSA 4 controls	>12 months	Picture–word matching	Perilesional activation in PSA, more widespread frontal and temporal activation than controls; reduced right hemisphere activation in PSA
[12]	82 PSA 82 controls	>6 months	Naming and semantic decision	No evidence of perilesional activation; normalization of the language network +/− activation of alternative cortical areas independent of distance from lesion
[13]	76 PSA 69 controls	>6 months	Semantic decision	Greater right hemisphere activation in PSA than controls, associated with higher education, younger age, left-handedness
[3]	14 PSA 14 controls	0–4 d *, 2 w, 4–12 months	Sentence judgment	Globally reduced activation acutely, then greater right hemisphere activation at 2–3 w, then normalization of language network activation in PSA
[14]	17 PSA	2, 6, 17, 26, 52 w	Semantic fluency	Increased left temporal, frontal, and cerebellar activation over time
[15]	8 PSA 14 controls	>6 months	Word stem completion	Less right cerebellar activation in PSA than controls. Increased right frontal and left cerebellar activation associated with learning in PSA

* d = days; w = weeks; m = months.

**Table 2 brainsci-15-00790-t002:** Studies of PSA that have compared average results across groups.

Reference	Number (N)	Time Since Stroke	Groups	Results
[16]	27	>1 year	Good (n = 9) vs. limited (n = 18) recovery	Greater left language network activation in good recovery; greater right homologous activation in PSA with limited recovery
[17]	34	<1 w *, 2–3 w, >6 m	Frontal (n = 17); temporoparietal (n = 17) lesions	Increased perilesional activation over time; right homologous activation only with frontal lesions
[18]	20	<3 m, 2 m later	Good (n = 9) vs. limited (n = 11) recovery	Connectivity between left fusiform and other left and right hemisphere language network increased in those with good recovery and decreased with limited recovery

* w = weeks; m = months.

## Data Availability

No new data were created or analyzed in this study.

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
