# Peer review of "Neural Bases of Language Recovery After Stroke Can Only Be Fully Understood Through Longitudinal Studies of Individuals"

_brainsci, 2025, doi:10.3390/brainsci15080790_

Round 1

Reviewer 1 Report

Comments and Suggestions for Authors

Professor Hillis addresses the highly relevant issue of how analysis of the restorative mechanisms underpinning recovery from post-stroke aphasia using longitudinal functional neuroimaging can be improved. Hillis is a pioneer in studying the mechanisms of neural plasticity implicated on post-stroke aphasia recovery using a sound neuroimaging methodology in both people with acute and chronic aphasia.  

Hillis reviews a hot topic in the neurorehabilitation of post-stroke aphasia and her article represents an important contribution to this research area. She proposes a paradigm change favoring the evaluation of single-case studies instead of group studies. The current Hillis position might be viewed as a back to future strategy (returning to the single-case study methodology popular at the end of the last century but steadily abandoned since then). Although I sympathize with her idea, I wonder whether researchers would adopt this strategy without objection.

Hillis points out that her proposal will permit dumping the current strategy of averaging functional imaging data across participants used in group studies. To stimulate increasing research by applying single-case studies methodology, Hillis provides guidelines on the design of future neuroimaging studies aimed at unveiling the neural correlations of aphasia recovery studied with neuroimaging. 

Her interesting and timely review article is divided into three parts. In each section, Hillis has selected representative examples of different testing modalities for longitudinal course of language recovery, explaining their strengths and weaknesses. The first section evaluates language function neuroimaging in neurotypical individuals, and it unveils consistent patterns of brain areas’ location that working together allows normal language functioning. The second part of this review is focused on the findings obtained with longitudinal functional neuroimaging in group studies on post-stroke aphasia. In this part, Hillis describes the usual disparity of results across studies which are reliant on the heterogeneity (e.g., lesion location and volume, demographic/clinical variables) of the studied populations. In the final section, Hillis emphasizes the need to perform longitudinal studies in individual patients and case series with post-stroke aphasia using functional neuroimaging. This strategy would control individual variability in the recovery process and will provide hints for the implementation of personalized treatments.

The aim of the manuscript is well defined. The conclusions reached by the author are relevant to readers of Brain Sciences and the content of the manuscript will be considered very useful by professionals working in the neural mechanisms underpinning post-stroke aphasia recovery. The Tables are clear, and their content does not overlap with the text. The figures are of good quality.

In summary, Hillis has provided a clear and provocative review article which will stimulate other researchers to move a step forward in the study of adaptive neural plasticity in post-stroke aphasia occurring spontaneously or promoted by treatment approaches.

Author Response

Comments 1: 

Professor Hillis addresses the highly relevant issue of how analysis of the restorative mechanisms underpinning recovery from post-stroke aphasia using longitudinal functional neuroimaging can be improved. Hillis is a pioneer in studying the mechanisms of neural plasticity implicated on post-stroke aphasia recovery using a sound neuroimaging methodology in both people with acute and chronic aphasia.  

Hillis reviews a hot topic in the neurorehabilitation of post-stroke aphasia and her article represents an important contribution to this research area. She proposes a paradigm change favoring the evaluation of single-case studies instead of group studies. The current Hillis position might be viewed as a back to future strategy (returning to the single-case study methodology popular at the end of the last century but steadily abandoned since then). Although I sympathize with her idea, I wonder whether researchers would adopt this strategy without objection.

Hillis points out that her proposal will permit dumping the current strategy of averaging functional imaging data across participants used in group studies. To stimulate increasing research by applying single-case studies methodology, Hillis provides guidelines on the design of future neuroimaging studies aimed at unveiling the neural correlations of aphasia recovery studied with neuroimaging. 

Her interesting and timely review article is divided into three parts. In each section, Hillis has selected representative examples of different testing modalities for longitudinal course of language recovery, explaining their strengths and weaknesses. The first section evaluates language function neuroimaging in neurotypical individuals, and it unveils consistent patterns of brain areas’ location that working together allows normal language functioning. The second part of this review is focused on the findings obtained with longitudinal functional neuroimaging in group studies on post-stroke aphasia. In this part, Hillis describes the usual disparity of results across studies which are reliant on the heterogeneity (e.g., lesion location and volume, demographic/clinical variables) of the studied populations. In the final section, Hillis emphasizes the need to perform longitudinal studies in individual patients and case series with post-stroke aphasia using functional neuroimaging. This strategy would control individual variability in the recovery process and will provide hints for the implementation of personalized treatments.

The aim of the manuscript is well defined. The conclusions reached by the author are relevant to readers of Brain Sciences and the content of the manuscript will be considered very useful by professionals working in the neural mechanisms underpinning post-stroke aphasia recovery. The Tables are clear, and their content does not overlap with the text. The figures are of good quality.

In summary, Hillis has provided a clear and provocative review article which will stimulate other researchers to move a step forward in the study of adaptive neural plasticity in post-stroke aphasia occurring spontaneously or promoted by treatment approaches.

Reply: Thank you for this insightful summary and critique.  I agree that some researchers would not adopt my proposed strategy without objection.  But I would challenge such researchers to defend the approach of averaging functional imaging results across patients with different lesions to draw conclusions that would be generalizable to any other stroke patients.  In the absence of conclusions that generalize to ANY other patients, I cannot see how such results enlighten the field.  I have altered the paper in response to other reviewers in ways that might make the proposed approach less objectionable. 

Reviewer 2 Report

Comments and Suggestions for Authors

Thank you for the opportunity to review this literature review and synthesis of brain imaging studies for post-stroke Aphasia (PSA). In the paper, the author argues that group-average studies cannot reliably or validly inform debates about how the language network recovers post-stroke as group averages obscure important individual variation for a number of variables.

I enjoyed reading this paper and I agree with the arguments made, I have a few suggestions for content and structure, and minor suggestions.

Comments for structure / concepts

*The line 193-194 "it is not justifiable to average functional imaging across individuals with stroke" and line 445 "reporting group averages will not help" - this is a nice, strong conclusion to draw, but then feels undercut by the paragraph in lines 420-430 that identifies useful group study data. I think some restructuring would help make this argument more robust.
*For example, what are the key differences in tasks and methods that make studies more or less useful? One theme that runs through is whether studies are looking at activation associated with improved scores (either longitudinally or across a group). This is an absolutely key argument and could be made more explicit throughout section 3 when discussing and critiquing studies. It would make sense to move section 420-430 in the Discussion to the end of Section 3, and compare explicitly these studies that are useful in understanding recovery to those summarised in Table 1 that give conflicting results.
*Linked to the above, the review paper by Hartwigsen and Saur (reference 19) could be moved earlier - to set up the subsequent review of literature (rather than coming at the end) and bringing the critiques from lines 181-190 and integrating them with the summary of literature in section 3. It would make the argument stronger if we can see the critiques presented alongside the summaries of papers rather than in this separate paragraph at the end.  
*Section 4 - this needs a short introductory paragraph contrasting the studies to be presented with those summarised in Section 3, and laying out explicitly why this approach with individuals is better. 
*Through section 4, there are a number of places where tasks are not described. For example, referring to 'language', 'language comprehension', 'word processing', 'word generation', 'task specific activation'. Please check through and change these to specific task descriptions, as found in Section 3 (e.g. auditory word comprehension, auditory sentence comprehension, written semantic judgement, spoken picture naming etc.).

*Line 196-198 "The average result for any given group..." - I agree wholeheartedly with this point. Another way to explore this issue is to look at variation directly, e.g. measures of variation such as SD or ranges, or plotting the individual variation explicitly. This has become more possible with mixed effect models of behavioural data that capture individual variation in random effects. I do not know how feasible that is with imaging data. How do the cited case-series manage and present data on individual variation? Is it only through presenting case by case data? Please describe this somewhere.
*Linked to the above, Conclusion section / Lines 455-457: can the author provide some guidance on what methods and data collection practices will help with robust design of case-series functional imaging data. Power is presumably a key issue, and the reason that group studies are common. How can this be tackled?

Minor comments / typos:
*Line 44: "Here I briefly review results of functional imaging of language"
*There are places where the acronym PSA is not used and the full term "post-stroke aphasia" is used instead. Please check through and use the acronym after it has been introduced in line 54.
*The term "normalization of language network" or similar is used in a few places. This needs defining - does it refer to re-activation of peri-lesional areas, or more neurotypical patterns or activation, or both? The first use is on Line 118 - can there be a definition here?
*Lines 85-90: please include references for the statement "individual neurotypical controls reliably show the same areas of activation associated with language as do groups, although longer imaging studies... are often required."
*Line 166 "A recent study used functional near infrared.." (delete that)
*Line 230 "They who showed that successful" - should be "They showed that"
*Line 315 "aphasia due stroke" - should be "aphasia due to stroke"
*Line 335 Figure 6 legend "bar on right" -  change to "The right hand color bar"
*Line 379-380 "This individual with global aphasia two months post-stroke involving language network" - change to "two months after a stroke that involved the language network"
*Line 401-402 "IFG during language in the left hemisphere" - give a specific task here, or define what 'during language' means. 

Author Response

Comments 1. *The line 193-194 "it is not justifiable to average functional imaging across individuals with stroke" and line 445 "reporting group averages will not help" - this is a nice, strong conclusion to draw, but then feels undercut by the paragraph in lines 420-430 that identifies useful group study data. I think some restructuring would help make this argument more robust.
*For example, what are the key differences in tasks and methods that make studies more or less useful? One theme that runs through is whether studies are looking at activation associated with improved scores (either longitudinally or across a group). This is an absolutely key argument and could be made more explicit throughout section 3 when discussing and critiquing studies. It would make sense to move section 420-430 in the Discussion to the end of Section 3, and compare explicitly these studies that are useful in understanding recovery to those summarised in Table 1 that give conflicting results.

Response 1.  I thank the reviewer for the many positive comments and helpful suggestions.  I have moved section 420-430 in the Discussion to the end of Section 3, as suggested, and compare explicitly these studies that are useful in understanding recovery to those summarized in Table 1. I add: A key point is that studies that evaluate correlations between improvement in scores and activation or connectivity might provide insights into the neural bases of recovery across individuals (likely only those with specific lesions and stages of stroke), whereas studies summarized in Table 1 do not provide clear insights about the recovery of any individual in the group or anyone else with PSA.  I add: "A key point is that studies that evaluate correlations between improvement in scores and activation or connectivity might provide insights into the neural bases of recovery across individuals (likely only those with specific lesions and stages of stroke), whereas studies summarized in Table 1 do not provide clear insights about the recovery of any individual in the group or anyone else with PSA."

Comment 2: *Linked to the above, the review paper by Hartwigsen and Saur (reference 19) could be moved earlier - to set up the subsequent review of literature (rather than coming at the end) and bringing the critiques from lines 181-190 and integrating them with the summary of literature in section 3. It would make the argument stronger if we can see the critiques presented alongside the summaries of papers rather than in this separate paragraph at the end.  

Response 2:  I did not move reference 19, because it is a review of aphasia recovery with various treatments (averaged across groups)- a wide area that I did not want to explicitly address, but wanted to acknowledge, as it is quite a comprehensive review.  I did carefully consider moving critiques presented alongside the summary of each study, but decided against it, because I did not have any specific critique any of the methods or conclusions of individual studies.  Rather, I wanted simply to describe the studies and results to point out the conflicting results and conclusions from studies that each individually seemed rational.  But the conclusions cannot all be correct as they come to very different conclusions about PSA recovery.  

Comment 3: *Section 4 - this needs a short introductory paragraph contrasting the studies to be presented with those summarised in Section 3, and laying out explicitly why this approach with individuals is better. 

Response 3: Thank you for this excellent suggestion.  I added an introductory paragraph at the beginning of Section 4: 

"Given the conflicting results obtained from group studies regarding changes in activation with recovery from PSA, depending on a variety of lesion and patient variables, the only clear solution is to evaluate changes in series of individuals of all types (ages, sexes, times post stroke, variety of lesion volumes and locations) to unveil the range of changes in the brain that support language recovery after stroke. In this section I review some of the published studies of functional imaging of individuals who have recovered from PSA, as well as longitudinal studies of individuals during PSA recovery. Many are from my own lab, as I have available the original images (not previously published) to illustrate distinct mechanisms of recovery. The studies certainly show heterogenous results, but do not draw conclusions meant to generalize across people with PSA. Rather, the aim is to show the range of mechanisms of recovery."

Comment 4: *Through section 4, there are a number of places where tasks are not described. For example, referring to 'language', 'language comprehension', 'word processing', 'word generation', 'task specific activation'. Please check through and change these to specific task descriptions, as found in Section 3 (e.g. auditory word comprehension, auditory sentence comprehension, written semantic judgement, spoken picture naming etc.).

Response 4: This point is well appreciated. I have added more specifics about the behavioral tasks in this section. Please see inserted text at lines 239-40, 243, 246-47, 248-50, 261-63, 274-75, 343-44, 362, 414-15, 427-28, 442, 450-51.

Comment 5: *Line 196-198 "The average result for any given group..." - I agree wholeheartedly with this point. Another way to explore this issue is to look at variation directly, e.g. measures of variation such as SD or ranges, or plotting the individual variation explicitly. This has become more possible with mixed effect models of behavioural data that capture individual variation in random effects. I do not know how feasible that is with imaging data. How do the cited case-series manage and present data on individual variation? Is it only through presenting case by case data? Please describe this somewhere.

Response 5: This is an excellent question.  I am not a statistician, but I address this (and the next comment) in a final paragraph, describing recommendations for amassing a data set with individual data (publicly available).  One could use intraclass correlation coefficients (ICC) to evaluate the agreement between 2 cases, or use nonparametric statistics to compare the frequency of occurrence of activation in particular areas.  However, once we accumulate a large data set of functional imaging of recovery from individuals, I am sure there will be very clever ways to describe the variation in activation (e.g. for one task vs. another; for one time post-stroke vs another) using AI.  Additionally, changes over time across individuals can be visualized using lasagna plots.  

Comment 6: *Linked to the above, Conclusion section / Lines 455-457: can the author provide some guidance on what methods and data collection practices will help with robust design of case-series functional imaging data. Power is presumably a key issue, and the reason that group studies are common. How can this be tackled?

Response 6: This is a suburb idea. I have added a final paragraph providing some guidance. I added at the end:

"What is needed is a collaboration of investigators who agree to bank data on a set of agreed upon imaging protocols as well as behavioral data, from individuals with PSA. Such large data sets have proven to be exceptionally useful for uncovering changes in the brain related to changes in behavior, such as the multitude of studies generated from the Alzheimer’s Disease Neuroimaging Initiative (https://adni.loni.usc.edu/). The Collaboration of Aphasia Trialists, an international network of aphasia investigators from numerous disciplines and from across more than 50 countries, has begun to discuss such an initiative focused on aphasia recovery (https://www.aphasiatrials.org/) Such a data set would provide the opportunity to use innovative statistics and artificial intelligence to describe and visualize the variations of recovery trajectories in the brain and in behavior, and how they are linked. For example, lasagna plots [43] can be used to visualize changes in language tasks across individuals over time. Although I would avoid averaging data across patients, with sufficient number of participants, non-parametric statistics can be used, for example, to evaluate associations between frequency or degree of activation or connectivity in particular areas and improvement on specific language tasks. But undoubtedly, biostatisticians will develop more sophisticated and clever ways to describe the variations in, and test hypotheses about, how brain activation changes in association with recovery of language."

Comments 7-16:  Minor comments / typos:

Comment 7. *Line 44: "Here I briefly review results of functional imaging of language"

Response 7: Corrected, thank you.

Comment 8: *There are places where the acronym PSA is not used and the full term "post-stroke aphasia" is used instead. Please check through and use the acronym after it has been introduced in line 54.

Response 8:  Completed as recommended.

Comment 9: *The term "normalization of language network" or similar is used in a few places. This needs defining - does it refer to re-activation of peri-lesional areas, or more neurotypical patterns or activation, or both? The first use is on Line 118 - can there be a definition here?

Response 9:  I defined it as "neurotypical patterns of activation in left inferior frontal, prefrontal, inferior parietal, and temporal regions" here.

Comment 10: *Lines 85-90: please include references for the statement "individual neurotypical controls reliably show the same areas of activation associated with language as do groups, although longer imaging studies... are often required."

Response 10: I added an appropriate reference.

Comment 11: *Line 166 "A recent study used functional near infrared.." (delete that)

Response 11: Corrected, thank you.

Comment 12: *Line 230 "They who showed that successful" - should be "They showed that"

Response 12: Corrected, thank you. 

Comment 13: *Line 315 "aphasia due stroke" - should be "aphasia due to stroke"

Response 13: Corrected, thank you.

Comment 14: *Line 335 Figure 6 legend "bar on right" -  change to "The right hand color bar"

Response 14: Edited as suggested.

Comment 15: *Line 379-380 "This individual with global aphasia two months post-stroke involving language network" - change to "two months after a stroke that involved the language network"

Response 15: Edited as suggested.

Comment 16: *Line 401-402 "IFG during language in the left hemisphere" - give a specific task here, or define what 'during language' means. 

Response 16: I added the specific task

Reviewer 3 Report

Comments and Suggestions for Authors

The manuscript offers an authoritative and insightful perspective on a highly relevant topic. With modest revisions to tone, framing, and clarity, it will serve as a valuable and influential contribution to the literature on post-stroke language recovery.

  1. The manuscript would benefit from a brief clarification of its review approach. Although clearly a narrative expert perspective, adding a short paragraph in the introduction to explain that this is a selective review based on decades of clinical and research experience would enhance transparency and guide reader expectations.
  2. While the central thesis advocating for longitudinal, individual-level imaging is compelling, several claims regarding the limitations of group-averaged studies are stated too categorically. For instance, phrases like “it is not justifiable to average functional imaging across individuals after stroke” may be softened slightly to maintain balance and scientific neutrality.
  3. The paper would be strengthened by a concise concluding paragraph that outlines key recommendations for future research or clinical application—e.g., standardization of longitudinal imaging protocols, or calls for large-scale individualized databases. Even in a personal review, such closing guidance adds practical value.

Author Response

Comment 1: The manuscript would benefit from a brief clarification of its review approach. Although clearly a narrative expert perspective, adding a short paragraph in the introduction to explain that this is a selective review based on decades of clinical and research experience would enhance transparency and guide reader expectations.

Response 1: Point well-taken. I added a brief clarification as recommended at the very beginning: "Here I provide a narrative review of functional imaging studies that pertain to language recovery after stroke, selected based on their relevance to the topic of neural mechanisms of aphasia recovery." 

Comment 2:  While the central thesis advocating for longitudinal, individual-level imaging is compelling, several claims regarding the limitations of group-averaged studies are stated too categorically. For instance, phrases like “it is not justifiable to average functional imaging across individuals after stroke” may be softened slightly to maintain balance and scientific neutrality.

Response 2: The review is meant to be provocative. Reviewer 2 liked the strong tone.  I retained the language about "not justifiable" to challenge readers to justify the practice. In the abstract I include the phrase, "On this basis, I argue that it is not justifiable..."

Comment 3: The paper would be strengthened by a concise concluding paragraph that outlines key recommendations for future research or clinical application—e.g., standardization of longitudinal imaging protocols, or calls for large-scale individualized databases. Even in a personal review, such closing guidance adds practical value.

Response 3: I completely agree. I added a paragraph at the end:

"What is needed is a collaboration of investigators who agree to bank data on a set of agreed upon imaging protocols as well as behavioral data, from individuals with PSA. Such large data sets have proven to be exceptionally useful for uncovering changes in the brain related to changes in behavior, such as the multitude of studies generated from the Alzheimer’s Disease Neuroimaging Initiative (https://adni.loni.usc.edu/). The Collaboration of Aphasia Trialists, an international network of aphasia investigators from numerous disciplines and from across more than 50 countries, has begun to discuss such an initiative focused on aphasia recovery (https://www.aphasiatrials.org/) Such a data set would provide the opportunity to use innovative statistics and artificial intelligence to describe and visualize the variations of recovery trajectories in the brain and in behavior, and how they are linked. For example, lasagna plots [44] can be used to visualize changes in language tasks across individuals over time. Although I would avoid averaging data across patients, with sufficient number of participants, non-parametric statistics can be used, for example, to evaluate associations between frequency or degree of activation or connectivity in particular areas and improvement on specific language tasks. But undoubtedly, biostatisticians will develop more sophisticated and clever ways to describe the variations in, and test hypotheses about, how brain activation changes in association with recovery of language."